# Effective Bud Induction of *Acacia mangium* and *A. auriculiformis* Without KNO_3_ and NH_4_NO_3_ in Media

**DOI:** 10.3390/plants14111720

**Published:** 2025-06-05

**Authors:** Lin Sun, Yanping Lu, Liejian Huang

**Affiliations:** Research Institute of Tropical Forestry, Chinese Academy of Forestry, Guangzhou 510520, China; sunlin721@163.com (L.S.); 15067100402@163.com (Y.L.)

**Keywords:** *Acacia*, tissue culture, nitrogen deficiency, bud induction, rooting, 6-BA

## Abstract

Stem segments of *Acacia mangium* and *A. auriculiformis* containing full axillary buds were used to study the effects of reduced amounts of the main nitrogen source in the growth media. This condition, referred to as nitrogen deficiency in this article and denoted as -N, involved the omission of ammonium nitrate and potassium nitrate from MS media, and its impact on bud induction was assessed. The results show that in media lacking nitrogen, the bud induction rate, contamination rate, browning rate, stem length, and leaf number of induced buds of *A. mangium* and *A. auriculiformis* varied depending on the different culture media used. The optimal bud induction medium for *A. mangium* and *A. auriculiformis* was as follows: 1/4MS (-N) + 1.0 mg·L^−1^ 6-BA + 0.2 g·L^−1^ chlorothalonil + 5 g·L^−1^ AGAR. The bud induction rates were 72.6% and 100.0%, respectively. There were no significant differences in the rooting rates of the induced buds between the -N treatment and the complete nutrient treatment. We found that the buds induced in the -N media did not show obvious symptoms of nitrogen deficiency, and their growth status was not significantly different from those induced in the complete nutrient media, which indicates that nitrogen is not essential for the bud induction of *A. mangium* and *A. auriculiformis*. The results of this study provide an important reference for conducting related research on other plants and have are greatly significant for the sustainable development of tissue culture technology in the future.

## 1. Introduction

*Acacia mangium* and *A. auriculiformis* were introduced in China in the 1960s [1,2]. Due to their economic benefits and ecological value (such as good adaptability, rapid growth, high yield, developed nodules, nitrogen fixation ability, long fibers, etc.), *A. mangium* and *A. auriculiformis* became important tree species for barren mountain afforestation, landscaping, and fuelwood in the south of China [3,4,5,6].

*Acacia* propagates through both sexual and asexual methods. Its seed progeny exhibit significant genetic differentiation in traits such as tree shape, resistance, wood properties, and other aspects. Cutting propagation of *Acacia* is greatly affected by the seasons and other factors. For example, excessively high temperatures in summer can enhance the respiratory function of cuttings, consume excessive nutrients, and also easily lead to bacterial infection at the cut ends, which is not conducive to root growth. At the same time, the rate of water evaporation is high, and the cuttings are prone to water loss and withering. As a result, to a certain extent, the plantation and utilization of *Acacia* species are limited [7,8]. Therefore, it is very important to establish efficient propagation technology to provide a large number of high-quality seedlings and meet the demands of *Acacia* plantation and utilization.

Tissue culture is an essential asexual propagation technology that has the advantages of controllable, convenient management, high reproductive efficiency, and the ability to maintain excellent parental traits [9]. In recent years, researchers have performed extensive research on *Acacia* tissue culture [10,11,12,13,14,15,16], which has supported the plantation and utilization of *Acacia*.

Using a suitable medium is the key to successfully establishing a tissue culture technique. Widely used media include the MS and improved MS media, which consist of a large number of elements, trace elements, iron salts, organic matter, and more. Each component plays an important role in the growth and development of plants. Among them, nitrogen is indispensable, serving as the basic component of chlorophyll, nucleic acid, protein, and more. It regulates many physiological processes in plants through signal transduction pathways, ultimately regulating plant growth and development [17,18,19]. During tissue culture, the exogenous nitrogen supplied in the medium comes from nitrate nitrogen (potassium nitrate) and ammonia nitrogen (ammonium nitrate) [20,21,22]. As potassium nitrate and ammonium nitrate are the main raw materials for the production of explosives, their purchase and use have been strictly controlled in most countries and regions in recent years, with China has also implementing strict supervision. They are classified as explosive, dangerous goods in the Identification of Major Hazard Sources of Hazardous Chemicals [23], which seriously impact their promotion and application in the tissue culture industry. Furthermore, few studies have examined the influence of nitrogen on *Acacia* tissue culture seedlings and the response to nitrogen in the process of *Acacia* tissue culture.

Therefore, in order to solve the above-mentioned problems, this study used *A. mangium* and *A. auriculiformis* as experimental materials to study the effect of nitrogen on bud induction. Through adjusting the proportion of MS medium and the concentration of exogenous cytokinin 6-BA, the optimal bud induction medium for *A. mangium* and *A. auriculiformis* under nitrogen-deficiency conditions was identified. In addition, the induced buds were subjected to a rooting culture to compare the differences in rooting between *A. mangium* and *A. auriculiformis* under nitrogen-deficient conditions and complete nutrient conditions. This evaluation of the effects of nitrogen-deficient bud induction lays the foundation for relevant research of *Acacia* nitrogen-deficiency culture. In addition, the research results will be very helpful for the sustainable development of tissue culture technology in the future.

## 2. Results

### 2.1. Effects of N and 6-BA on Bud Induction

Under the -N treatment, the bud induction rate and browning rate of *A. mangium* were not significantly different between the different media used. The bud induction rate decreased with the increase in the concentration of the basic medium. The bud induction rates, from highest to lowest, were 1/8MS = 1/4MS > 1/2MS > MS, with the highest induction rate reaching 66.9%. The browning rates in all media used were lower than 20%, and the contamination rates were lower than 5%. The contamination rate in the 1/4MS medium was significantly higher than that in the 1/2MS medium (Figure 1).

Under the F treatment, with the increase in the concentration of the basic medium, the bud induction rates of *A. mangium* decreased significantly, and the browning rates increased significantly. The bud induction rates, from highest to lowest, were 1/8MS > 1/4MS > 1/2MS > MS, with the highest induction rate reaching 88.9%. The browning rates, from highest to lowest, were MS > 1/2MS > 1/4MS > 1/8MS, with the highest browning rate reaching 78.5%. The contamination rates in all media were lower than 10%, and the contamination rate in the 1/2MS medium was significantly higher than that in the 1/4MS medium and 1/8MS medium (Figure 1).

Under the -N treatment, with the increase in the concentration of the basic medium, the bud induction rates of *A. auriculiformis* increased first and then decreased. The bud induction rate in the 1/4MS medium was the highest, at 93.8%, and the induction rate in the MS medium was significantly lower than that in the other treatments. There were no significant differences in the browning rate or contamination rate of the different media (Figure 2).

Under the F treatment, with the increase in the concentration of the basic medium, the bud induction rates of *A. auriculiformis* decreased significantly, and the browning rates increased significantly. The bud induction rates, from highest to lowest, were 1/8MS > 1/4MS > 1/2MS > MS, with the highest induction rate reaching 88.6%. The browning rates, from highest to lowest, were MS > 1/2MS > 1/4MS >1/8MS, with the highest browning rate reaching 55.6%. There were no significant differences in the contamination rates in the different media, which were all lower than 5% (Figure 2).

Under the different concentrations of 6-BA, there were no significant differences in the bud induction rates, browning rates, or contamination rates of *A. mangium* and *A. auriculiformis* between the nitrogen-deficient media and complete nutrient media. Under the same 6-BA treatment concentration, the bud induction rates of *A. mangium* and the bud induction rates and contamination rates of *A. auriculiformis* were all higher in the nitrogen-deficient media than in the complete nutrient media. Conversely, the browning rates and contamination rates of *A. mangium* and the browning rates of *A. auriculiformis* were all higher in the complete nutrient media than in the nitrogen-deficient media (Figure 1 and Figure 2).

### 2.2. Effects of N on Stem Length and Leaf Number of Induced Buds

The results of multiple comparative analysis show that the lack of nitrogen in the media had a certain effect on the bud induction of *A. mangium* and *A. auriculiformis*. Among the 12 -N treatments, *A. mangium* and *A. auriculiformis* had the highest bud induction rates with treatment 8- (-N), with 72.6% and 100.0%, respectively. Among the 12 F treatments, the bud induction rate of *A. mangium* was the highest in treatment 10- (F), with 91.9%; the bud induction rate of *A. auriculiformis* was the highest in treatment 12- (F), with 94.1%. The highest bud induction rate of *A. auriculiformis* treated with -N was higher than that of the F treatment. The stem lengths of the induced *A. mangium* and *A. auriculiformis* buds in the complete nutrient media were higher than those in the nitrogen-deficient media (Table 1), indicating that suitable media can normally induce germination of *Acacia* macrophylla explants under nitrogen-deficiency conditions, but it will affect the elongation and growth of the buds to some extent. After 40 days, it was observed that the number of leaves on buds induced in the nitrogen-deficient culture was lower than those induced in the complete nutrient culture, and the leaf color was light green; however, no symptoms of nitrogen deficiency were observed (Figure 3). These results indicate that the bud induction rate and bud growth of *A. mangium* and *A. auriculiformis* were not significantly negatively affected under the absence of nitrogen. Comprehensive analysis showed that the optimal nitrogen-deficient medium for both *A. mangium* and *A. auriculiformis* bud induction was 1/4MS (-N) + 1.0 mg·L^−1^ 6-BA.

### 2.3. Rooting of Buds Induced in -N and F Treatments

*A. mangium* and *A. auriculiformis* buds induced under the -N and F treatments were subsequently cultured for rooting. White root primordia began to appear at the bottom of the stem at around 10 d, and fine roots began to appear at around 15 d. The root growth tended to be stable after one month, and the number of fibrous roots on the buds induced under the -N treatment was lower than those induced under the F treatment (Figure 4 and Figure 5). Different concentrations of IBA and NAA had significant effects on the rooting of *A. mangium* and *A. auriculiformis*. For the rooting culture of buds induced under the -N treatment, the combined concentrations of 0.5 mg·L^−1^ NAA and 1.0–2.0 mg·L^−1^ IBA was more favorable for the rooting of *A. mangium* under nitrogen-deficient culture. Treatment 8 (2.0 mg·L^−1^ IBA + 0.5 mg·L^−1^ NAA) led to a significantly higher rooting rate than the other treatments, with the highest rooting rate reaching 77.8% (Table 2). When the concentration of NAA was 0.1 mg·L^−1^, the rooting rate of *A. auriculiformis* gradually increased with the increase in the IBA concentration. Treatment 4 (2.0 mg·L^−1^ IBA + 0.1 mg·L^−1^ NAA) achieved the highest rooting rate, which was 88.9% (Table 2). For the rooting culture of buds induced under the F treatment, the rooting rates of the *A. mangium* buds under treatment 5 (0.5 mg·L^−1^ IBA + 0.5 mg·L^−1^ NAA) and treatment 12 (2.0 mg·L^−1^ IBA + 1.0 mg·L^−1^ NAA) were significantly higher than those of the other treatments. The rooting rate under treatment 12 was the highest, reaching 80.0% (Table 2). The rooting rate of the *A. auriculiformis* buds under treatment 3 (1.5 mg·L^−1^ IBA + 0.1 mg·L^−1^ NAA) was significantly higher than those of the other treatments, with the highest rooting rate reaching 90.0% (Table 2). The seedlings of *A. mangium* and *A. auriculiformis* rooted in the nitrogen-deficient media exhibited green leaves, robust growth, and no symptoms of nitrogen deficiency. Additionally, there was no significant difference compared to the seedlings rooted in the complete nutrient media (Figure 4 and Figure 5).

## 3. Discussion

In this study, the effects of nitrogen deficiency on the bud induction of *A. mangium* and *A. auriculiformis* were studied. The research results show that nitrogen deficiency reduced the stem length and leaf number of the induced buds, but the induced buds grew robustly with dark green leaves and did not show obvious deficiency symptoms. *A. mangium* had the highest bud induction rates when treated with 1/4MS (-N) + 1.0 mg·L^−1^ 6-BA and 1/8MS (F) + 0.5 mg·L^−1^ 6-BA, reaching 72.6% and 91.9%, respectively, with no significant difference between the treatments with highest bud induction rates. *A. auriculiformis* had the highest bud induction rates when treated with 1/4MS (-N) + 1.0 mg·L^−1^ 6-BA and 1/8MS (F) + 1.5 mg·L^−1^ 6-BA, reaching 100.0% and 94.1%, respectively, with no significant difference between the treatments with highest bud induction rates. The highest rooting rates of the *A. mangium* buds induced under the -N and F treatments were 77.8% and 80.0%, respectively, with no significant difference between them. The highest rooting rates of the *A. auriculiformis* buds induced under the -N and F treatments were 88.9% and 90.0%, respectively, with no significant difference between them. Under the nitrogen-deficiency conditions, *A. mangium* and *A. auriculiformis* buds could be induced and grow normally, indicating that nitrogen is not essential during bud induction.

Different plants need different nutrition for growth and development during tissue culture. This study showed that *A. mangium* and *A. auriculiformis* buds could be induced in both nitrogen-deficient and complete nutrient media, indicating that nitrogen had little influence on bud induction. The degree of explant browning is directly related to the synthesis of phenolic substances [24]. An excessive concentration of inorganic salts in the medium may cause explants to produce a large amount of phenolic substances, with then undergo oxidation reactions, leading to browning [25,26]. This study showed that the inducted buds of *A. mangium* and *A. auriculiformis* were affected by the concentration of MS medium; the higher concentration ratio, the lower the bud induction rate and the higher the browning rate. This indicates that the bud induction of *Acacia* was very sensitive to high concentrations of the complete nutrient medium, which may have been due to the inhibition of germination caused by the penetration and toxicity of high concentrations of inorganic salts. In the nitrogen-deficient media, the browning rates of *A. mangium* and *A. auriculiformis* were lower than those in the complete nutrient media, and the bud induction rates in the MS (-N) and 1/2MS (-N) media were significantly higher than those in the corresponding complete nutrient media. This indicates that the lack of ammonium nitrate and potassium nitrate in the medium reduced the content of inorganic salts and inhibited the oxidation of phenolic substances in the explants. Therefore, the browning of the explants decreased, mortality was reduced, and the bud induction rates significantly increased.

The appropriate concentration of growth regulators can promote the induction and growth of buds. Qiu et al. found that adding appropriate plant hormones could effectively promote cell growth and alleviate browning [27]; Feng et al. found that the browning rate of explants increased with increases in the 6-BA concentration [28]. In this study, there were no significant differences in the browning rate, contamination rate, or bud induction rate of *A. mangium* and *A. auriculiformis* when the concentration of 6-BA was in the range of 0.5–1.5 mg·L^−1^. This indicates that different plants have different tolerance degrees to the concentration of 6-BA, but 6-BA had no effect on the bud induction of *A. mangium* or *A. auriculiformis*. Growth regulators are crucial for the rooting of seedlings. Different concentrations of IBA and NAA have been used in the rooting of *Acacia* in vitro. We compared the buds induced in nitrogen-deficient and complete nutrient media by culturing them in rooting media with certain concentrations of IBA and NAA. The rooting rates ranged from 20% to 90%, depending on the combined concentrations of the growth media. *A. mangium* and *A. auriculiformis* had higher rooting rates in both the nitrogen-deficient and complete nutrient media with suitable concentrations of IBA and NAA, which further shows that nitrogen is not essential for *Acacia* tissue culture.

Under nitrogen stress, plants will show certain resistance. A study showed that in response to nitrogen-deficiency stress, *Hordeum vulgare* var. coeleste L. adapted by changing the activities of POD, CAT, and APX. These changes reduced the damage caused by nitrogen-deficiency stress, together playing a key role in enabling *Hordeum vulgare* var. coeleste L. to adapt to such stress [29]. Huo et al. found that when apple leaves and roots were under nitrogen-deficient conditions, the *MDATG*10 gene was overexpressed, which in turn regulated the absorption and transformation of nitrogen, reduced the impact of nitrogen stress on chloroplasts, and enabled the plants to grow normally [30]. This study found that buds of *A. mangium* and *A. auriculiformis* could be effectively induced when a certain concentration of 6-BA was added to the media. Additionally, the bud induction of *A. auriculiformis* increased the demand for essential nutrients under the conditions of nitrogen deficiency. We speculate that under nitrogen-deficiency stress, explants of *A. auriculiformis* require more nutrients and stimulation, while *A. mangium* explants are more affected by 6-BA. *A. mangium* and *A. auriculiformis* adjust the absorption and transformation of nitrogen through their own physiological adaptation to nitrogen-deficient environments to meet their normal growth needs.

In summary, *A. mangium* and *A. auriculiformis* tissue culture seedlings were able to achieve normal growth and metabolism under nitrogen-deficient conditions. The buds induced under the nitrogen-deficiency treatment did not show obvious symptoms of nitrogen deficiency, and their growth status was not very different from those induced under the complete nutrient treatment. Additionally, the buds induced under nitrogen deficiency could root normally, indicating that nitrogen is not essential in the tissue culture of *A. mangium* and *A. auriculiformis*. This study conducted a rooting experiment on buds induced in nitrogen-deficient medium, elucidating important condition requirements for growth and development in subsequent seedling refining and transplanting processes. It also successfully solved the challenge posed by the difficulty of purchasing potassium nitrate and ammonium nitrate for tissue culture production of *A. mangium* and *A. auriculiformis,* while providing a new idea for the sustainable development of tissue culture technology in the future.

## 4. Materials and Methods

### 4.1. Locations and Materials

The experimental materials were collected from the cutting orchard of *A. mangium* and *A. auriculiformis* at the *Acacia* Breeding Base in Shadui Town, Jiangmen City, Guangdong Province. The selected new shoots of the current year of *A. mangium* and *A. auriculiformis* exhibited good growth and were free from pests and diseases.

### 4.2. Methods

#### 4.2.1. Pretreatment of Explants

The selected new shoots of *A. mangium* and *A. auriculiformis* were soaked in washing powder dilution for 30 min and then rinsed. Then, a soft-bristled brush dipped in washing powder dilution was used to clean the surfaces of the branches. The stems with leaves were cut into 3–5 cm segments, retaining 1/3 of the original leaves and containing full axillary buds, to be used as explants. The pruned stems were rinsed with running water for more than 1 h, and then completely immersed in 0.8 g·L^−1^ carbendazim solution for disinfection for 3 min [31,32]. Then, the explants were rinsed with tap water, and filter paper was used to dry the surface of the explants before inoculation.

#### 4.2.2. Screening of Nitrogen-Deficient Medium

The explants were inoculated on MS (-N), 1/2MS (-N), 1/4MS (-N), and 1/8MS (-N), where -N indicates that the culture medium lacked the main nitrogen sources ammonium nitrate and potassium nitrate. Then, different concentrations of 6-BA (0.5 mg·L^−1^, 1.0 mg·L^−1^, and 1.5 mg·L^−1^) were added to the media, followed by 5 g·L^−1^ AGAR (AisaBio, Gel strength > 1400 g/cm^2^) and 0.2 g·L^−1^ chlorothalonil. Each treatment was controlled with the same proportion of complete nutrient media, marked as F (the treatment media are shown in Table 3). Each treatment was inoculated with three explants per bottle, for a total of 15 bottles, and was repeated three times. The growth status of the explants was observed and recorded. After 40 days of culture, the following bud induction-related indicators were recorded: bud induction rate, contamination rate, browning rate, bud length, and leaf number.

#### 4.2.3. Rooting Culture

The buds induced in the nitrogen-deficient medium and complete nutrient medium were inoculated into rooting medium for rooting culture. The rooting medium consisted of MS medium with different concentrations of IBA (0.5 mg·L^−1^, 1.0 mg·L^−1^, 1.5 mg·L^−1^, and 2.0 mg·L^−1^) and NAA (0.1 mg·L^−1^, 0.5 mg·L^−1^, and 1.0 mg·L^−1^). A total of 12 treatments were set up. Each treatment was inoculated with three strains per bottle, for a total of 15 bottles, and repeated three times. After 30 days, rooting was observed, and the rate of rooting was calculated.

### 4.3. Culture Conditions

The media did not require high-temperature autoclave sterilization, and the operating environment was not strictly sterile. The culture conditions were as follows: pH of 5.8 ± 0.2, temperature of (25 ± 2) °C, 12 h of light per day, and light intensity of 2500 Lux.

### 4.4. Data Processing

Excel was used to organize the collected data, and the SPSS 26.0 statistical analysis software was used for analysis of variance (*p* < 0.05).

## Figures and Tables

**Figure 1 plants-14-01720-f001:**
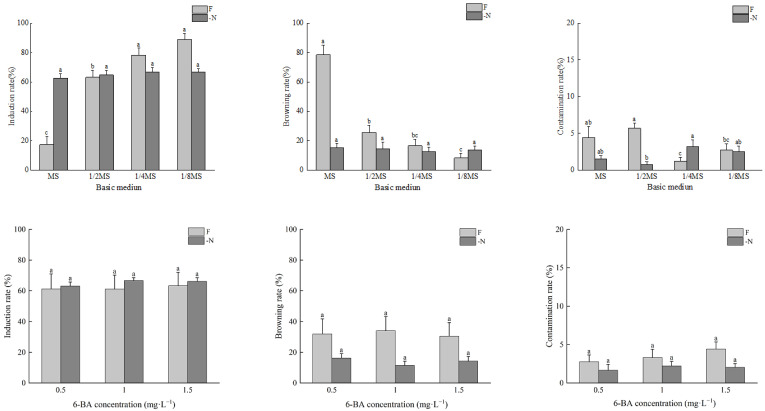
The effects of N and 6-BA on bud induction of *A. mangium.* Different lower-case letters indicate significant differences at *p* < 0.05.

**Figure 2 plants-14-01720-f002:**
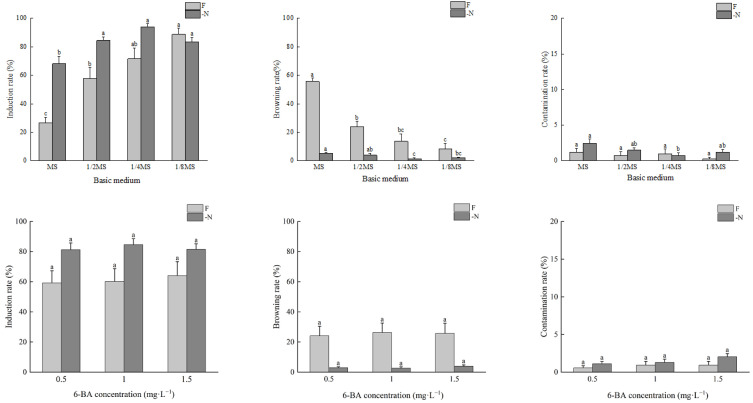
The effects of N and 6-BA on bud induction of *A. auriculiformis.* Different lower-case letters indicate significant differences at *p* < 0.05.

**Figure 3 plants-14-01720-f003:**
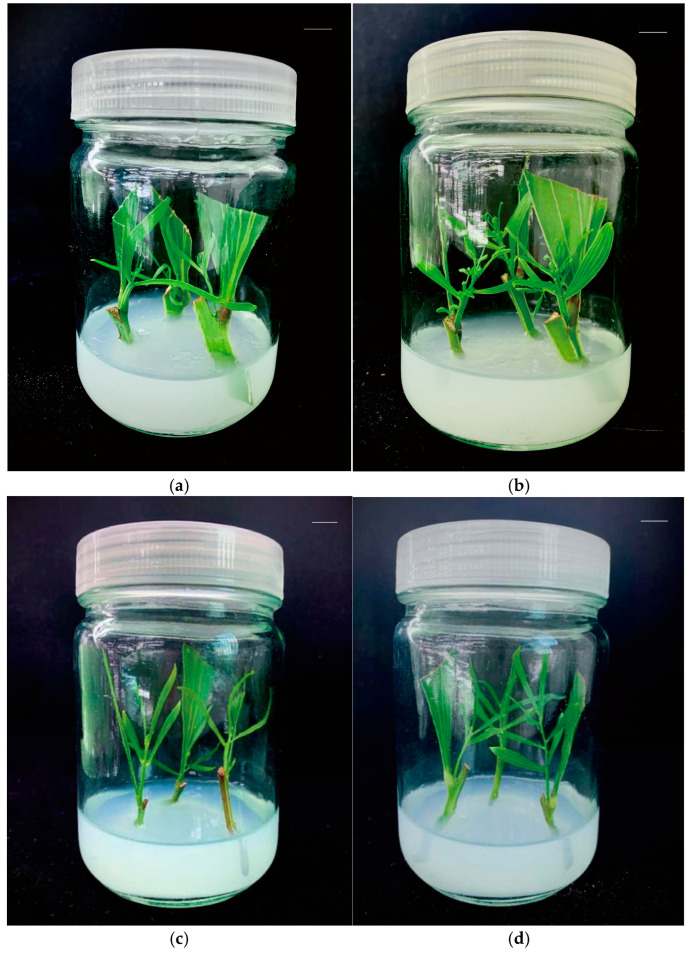
*A. mangium-* and *A. auriculiformis*-induced buds. (**a**) *A. mangium* buds induced in treatment 8- (-N). (**b**) *A. mangium* buds induced in treatment 10- (F). (**c**) *A. auriculiformis* buds induced in treatment 8- (-N). (**d**) *A. auriculiformis* buds induced in treatment 12- (F). Scale bars (top right-hand corner in each picture) = 1 cm.

**Figure 4 plants-14-01720-f004:**
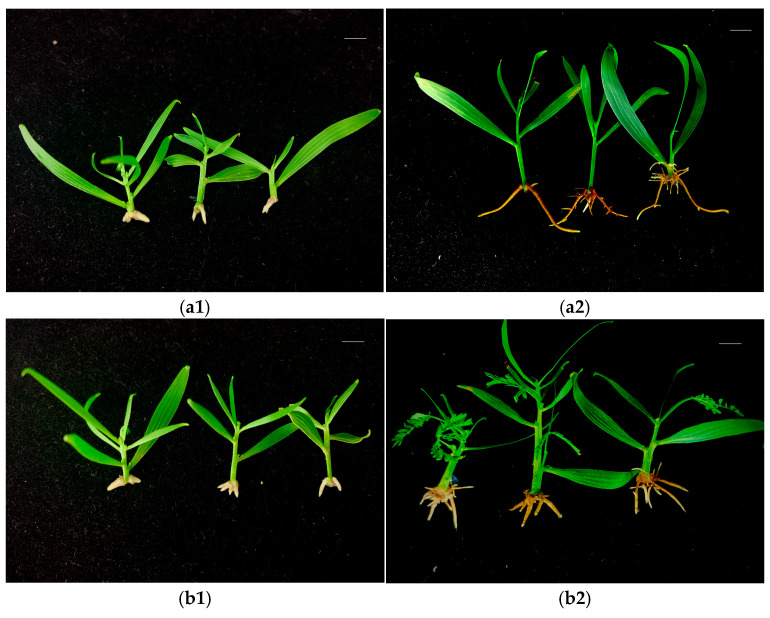
The rooting of *A. mangium*-induced buds. (**a1**) -N treatment-induced bud rooting after 15 days. (**a2**) -N treatment-induced bud rooting after 30 days. (**b1**) F treatment-induced bud rooting after 15 days. (**b2**) F treatment-induced bud rooting after 30 days. Scale bars (top right-hand corner in each picture) = 1 cm.

**Figure 5 plants-14-01720-f005:**
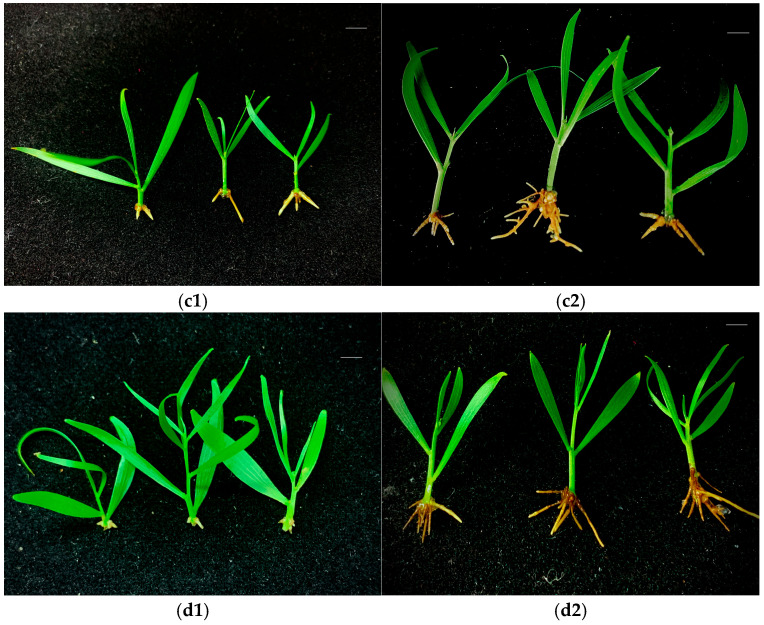
The rooting of *A. auriculiformis*-induced buds. (**c1**) -N treatment-induced bud rooting after 15 days. (**c2**) -N treatment-induced bud rooting after 30 days. (**d1**) F treatment-induced bud rooting after 15 days. (**d2**) F treatment-induced bud rooting after 30 days. Scale bars (top right-hand corner in each picture) = 1 cm.

**Table 1 plants-14-01720-t001:** The stem lengths and leaf numbers of induced *A. mangium* and *A. auriculiformis* buds.

No.	Treatments	*A. mangium*			*A. auriculiformis*		
		Bud Induction Rate (%)	Stem Length (mm)	Leaf Number	Bud Induction Rate (%)	Stem Length (mm)	Leaf Number
1	MS (-N), 0.5 mg/L 6-BA	65.2 b	7.0 b	2.5 b	64.4 b	7.2 b	2.2 b
	MS (F), 0.5 mg/L 6-BA	12.6 c	7.1 b	2.5 b	29.6 c	8.9 b	2.1 b
2	MS (-N), 1.0 mg/L 6-BA	66.7 b	8.2 b	2.9 b	70.4 b	8.3 b	2.7 b
	MS (F), 1.0 mg/L 6-BA	17.1 c	8.8 b	2.6 b	27.4 c	11.2 ab	2.5 b
3	MS (-N), 1.5 mg/L 6-BA	56.3 b	8.1 b	3.1 ab	69.6 b	8.4 b	2.8 b
	MS (F), 1.5 mg/L 6-BA	22.2 c	8.5 b	3.0 b	23.0 c	11.1 ab	2.8 b
4	1/2MS (-N), 0.5 mg/L 6-BA	64.5 b	8.2 b	3.0 ab	83.0 ab	8.2 b	2.2 b
	1/2MS (F), 0.5 mg/L 6-BA	61.5 b	11.5 ab	3.7 ab	55.5 bc	12.4 ab	2.9 b
5	1/2MS (-N), 1.0 mg/L 6-BA	60.7 b	8.5 b	3.2 ab	83.0 ab	9.3 b	2.9 b
	1/2MS (F), 1.0 mg/L 6-BA	60.7 b	15.0 a	4.7 a	57.1 b	14.6 a	3.4 ab
6	1/2MS (-N), 1.5 mg/L 6-BA	68.9 b	9.2 b	3.6 ab	87.4 ab	9.8 b	2.6 b
	1/2MS (F), 1.5 mg/L 6-BA	67.4 b	14.4 ab	4.6 ab	60.7 b	14.7 a	3.7 ab
7	1/4MS (-N), 0.5 mg/L 6-BA	60.0 b	7.4 b	2.8 b	95.5 ab	8.9 b	2.4 b
	1/4MS (F), 0.5 mg/L 6-BA	79.3 ab	14.1 ab	4.5 ab	66.7 b	13.4 ab	3.1 ab
8	1/4MS (-N), 1.0 mg/L 6-BA	72.6 ab	8.5 b	3.1 ab	100.0 a	9.8 b	2.8 b
	1/4MS (F), 1.0 mg/L 6-BA	77.8 ab	14.5 ab	4.4 ab	69.6 b	13.8 ab	3.4 ab
9	1/4MS (-N), 1.5 mg/L 6-BA	68.1 b	7.9 b	3.4 ab	85.9 ab	10.5 ab	3.0 ab
	1/4MS (F), 1.5 mg/L 6-BA	77.8 ab	14.1 ab	4.4 ab	78.5 ab	14.0 ab	3.8 a
10	1/8MS (-N), 0.5 mg/L 6-BA	63.0 b	7.8 b	2.9 b	82.2 ab	9.5 b	2.6 b
	1/8MS (F), 0.5 mg/L 6-BA	91.9 a	13.4 ab	4.3 ab	85.2 ab	13.6 ab	3.2 ab
11	1/8MS (-N), 1.0 mg/L 6-BA	66.6 b	7.8 b	3.2 ab	85.2 ab	9.4 b	2.5 b
	1/8MS (F), 1.0 mg/L 6-BA	88.9 ab	13.8 ab	4.3 ab	86.7 ab	14.7 a	3.4 ab
12	1/8MS (-N), 1.5 mg/L 6-BA	71.1 ab	9.0 b	3.5 ab	83.0 ab	10.3 ab	2.8 b
	1/8MS (F), 1.5 mg/L 6-BA	85.9 ab	14.6 ab	4.3 ab	94.1 ab	14.8 a	3.7 ab

Different lower-case letters indicate significant differences at *p* < 0.05. Data are means ± standard error.

**Table 2 plants-14-01720-t002:** The rooting rates of induced *A. mangium* and *A. auriculiformis* buds.

No.	Treatment		*A. mangium*		*A. auriculiformis*	
	Concentration of IBA (mg/L)	Concentration of NAA (mg/L)	Rooting Rate (-N)	Rooting Rate (F)	Rooting Rate (-N)	Rooting Rate (F)
1	0.5	0.1	33.3 cd	41.1 bcd	66.7 bcd	58.9 cd
2	1.0		24.4 cd	34.4 cd	76.7 ab	75.6 abc
3	1.5		28.9 cd	52.2 bc	80.0 ab	90.0 a
4	2.0		31.1 cd	33.3 acd	88.9 a	64.5 bcd
5	0.5	0.5	21.1 d	75.6 a	54.4 cd	52.2 d
6	1.0		58.9 b	31.1 d	50.0 d	66.7 bcd
7	1.5		66.7 ab	47.8 bcd	54.5 cd	60.0 cd
8	2.0		77.8 a	56.7 b	64.4 bcd	51.1 d
9	0.5	1.0	37.8 c	47.8 bcd	57.8 cd	54.4 d
10	1.0		34.4 cd	57.8 b	66.7 bcd	61.1 cd
11	1.5		34.4 cd	46.7 bcd	82.2 a	65.5 bcd
12	2.0		38.9 c	80.0 a	71.1 bc	78.9 abc

Different lower-case letters indicate significant differences at *p* < 0.05. Data are means ± standard errors.

**Table 3 plants-14-01720-t003:** Treatment media used in this study.

No.	Basic Medium	Concentration of 6-BA (mg/L)
1	MS(-N), MS(F)	0.5
2	MS(-N), MS(F)	1.0
3	MS(-N), MS(F)	1.5
4	1/2MS(-N), 1/2MS(F)	0.5
5	1/2MS(-N), 1/2MS(F)	1.0
6	1/2MS(-N), 1/2MS(F)	1.5
7	1/4MS(-N), 1/4MS(F)	0.5
8	1/4MS(-N), 1/4MS(F)	1.0
9	1/4MS(-N), 1/4MS(F)	1.5
10	1/8MS(-N), 1/8MS(F)	0.5
11	1/8MS(-N), 1/8MS(F)	1.0
12	1/8MS(-N), 1/8MS(F)	1.5

## Data Availability

Data sharing is not applicable to this article.

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
