# Peer review of "Effective Bud Induction of Acacia mangium and A. auriculiformis Without KNO3 and NH4NO3 in Media"

_plants, 2025, doi:10.3390/plants14111720_

Round 1
Reviewer 1 Report
Comments and Suggestions for Authors
Line 10: The claim that buds showed "no obvious symptoms of nitrogen deficiency" lacks biochemical validation. Include quantitative data such as chlorophyll content or nitrogen concentration measurements to support visual observations.
Line 47: The statement about cutting propagation being "greatly affected by seasons" is vague. Cite specific environmental factors (e.g., temperature, humidity) influencing seasonal variability.
Line 60: The assertion that nitrogen is "indispensable" contradicts findings of successful bud induction without nitrogen. Clarify whether nitrogen is dispensable only during induction or throughout growth.
Line 85: No justification is provided for omitting autoclaving. Chlorothalonil suppresses fungi but not bacteria; include microbial load data to validate contamination control.
Line 127: The claim "1/8MS=1/4MS>1/2MS>MS" lacks statistical significance markers. Report p-values or confidence intervals for comparisons.
Lines 146-155 (Table 2): Stem length/leaf number differences are noted but not contextualized. Discuss how shorter stems might impact rooting efficiency or field performance.
Lines 179-204 (Figures 3-5): Missing images make claims about growth differences unverifiable. Include high-resolution images with scale bars.
Line 217: Speculation about "physiological adaptation to nitrogen deficiency" lacks mechanistic evidence. Measure stress markers (e.g., proline, antioxidant enzymes) to support claims.
Line 230: Reliance on outdated references (e.g., Skolmen 1986) overlooks modern nitrogen metabolism studies. Update citations to include recent work (e.g., post-2015).
Line 72: The term "washing powder dilution" is ambiguous. Define concentration (e.g., 2% w/v) and specify the brand for reproducibility.
Line 134: The statement "highest induction rate was 93.8%" lacks statistical context. Add ANOVA/Tukey test results to clarify significance.
Line 156: Describing leaf color as "light green" is subjective. Use spectrophotometric chlorophyll measurements for objectivity.
Line 240: The claim that plants "adjust nitrogen absorption" lacks supporting data. Include measurements of N-uptake efficiency or transporter gene expression.
Line 210: Asserting "no significant difference" in rooting rates without statistical proof is incomplete. Report p-values in tables.
Line 68: Retaining "1/3 of original leaves" is arbitrary. Justify this ratio with references or preliminary experiments.
Line 350 (References): Song et al. (2021) on chemical safety is tangential. Replace with tissue culture-specific sterilization literature.
Line 270: Failure to discuss implications of "shorter stems/fewer leaves" leaves a gap. Address how this morphology affects acclimatization or survival.
Line 42 (Keywords): Missing terms like "cytokinin" or "6-BA" reduces discoverability. Include these keywords.
Line 142: Attributing contamination rate differences to media without microbial analysis is incomplete. Test explant microbial load for clarity.
Line 350 (Author Contributions): Roles like "review and editing" are vague. Specify contributions (e.g., "Huang designed experiments").
Line 196: Rooting images lack scale bars, preventing assessment of root quality. Add scale bars.
Line 72: Mentioning China’s "key supervision" of nitrates is unsubstantiated. Cite specific regulations (e.g., national safety codes).
Lines 216-225 (Table 3): Rooting rates lack mention of biological replication. Clarify if experiments were repeated across seasons.
Not discussing enough nitrogen-fixing symbiosis in Acacia misses a key trait. Address whether nodulation compensates for N-deficiency in vitro.
Comments on the Quality of English LanguageThe English is generally understandable but requires thorough editing for clarity, precision, and professionalism. Minor revisions to grammar, terminology, and phrasing will enhance readability and align the manuscript with academic standards.
Reviewer 2 Report
Comments and Suggestions for Authors
The manuscript investigates the feasibility of inducing axillary buds from stem segments of Acacia mangium and A. auriculiformis using MS-based media lacking ammonium nitrate and potassium nitrate (termed -N or nitrogen deficiency). The study compares various dilutions of this -N medium with corresponding complete (F) media, supplemented with different 6-BA concentrations. Rooting of buds induced from optimal -N and F media is also assessed. The authors conclude that nitrogen (specifically NH₄NO₃ and KNO₃) is not essential for bud induction and initial rooting in these species under their experimental conditions, potentially offering a solution to procurement issues with these chemicals. While the findings are potentially interesting for tissue culture practice, the manuscript has several limitations regarding design, reporting, and interpretation.
- Definition of "-N": The term "nitrogen deficiency (-N)" is used throughout but specifically refers only to the omission of NH₄NO₃ and KNO₃ from MS medium. MS medium contains other nitrogen sources (e.g., in amino acids like glycine, if added as standard, though not explicitly mentioned here; potential trace contaminants). While NH₄NO₃ and KNO₃ are the major N sources, claiming complete "nitrogen deficiency" is inaccurate. A more precise term like "Major N Source Omission Medium" or similar might be better. The rationale for removing both major N sources, rather than testing effects of removing only NH₄⁺ or only NO₃⁻, or reducing total N, is not provided.
- Explant Source and Preparation: Explants were taken from a cutting orchard. Details on the age, physiological status, or specific genotype/clone of the source plants are missing. Variability between source plants could influence results. The pretreatment uses washing powder, which is non-standard and could introduce confounding chemical factors; standard surface sterilization protocols (e.g., ethanol, sodium hypochlorite) are typically used after initial washing. Carbendazim is used, but only briefly mentioned.
- Lack of Sterilization: The methods state "The media do not need to high temperature autoclave sterilization, and the operating environment is not strictly sterile". This is a highly unconventional approach for plant tissue culture, which typically demands strict sterility to prevent microbial contamination. While chlorothalonil is added, its effectiveness against the full spectrum of potential contaminants in a non-sterile environment is questionable. The reported low contamination rates (<10% generally) under these conditions are surprising and require strong justification or evidence that contamination was rigorously assessed (e.g., microscopic examination, plating out). This non-standard approach severely limits the replicability and comparability of the study.
- Media Composition: The study focuses on removing major N salts but doesn't specify the exact composition of the modified MS media (e.g., were other salts adjusted to compensate for removed K⁺? Was glycine included?). The type/brand of MS basal salt mix used isn't stated. AGAR concentration is 5 g/L, which is lower than typical (usually 6-8 g/L); justification is needed.
- Rooting Experiment Design: Buds from the "optimal" -N medium (Treatment 8: 1/4 MS(-N) + 1.0 6-BA) were compared to buds from multiple F media (specifically Tx 10-F for A. mangium and Tx 12-F for A. auriculiformis based on highest induction rate, not necessarily optimal growth). A more direct comparison would be rooting buds from Tx 8-(-N) vs. Tx 8-(F). Furthermore, the rooting media themselves contained complete MS nutrients, meaning the "-N" buds were transferred to N-replete conditions for rooting, which complicates interpretation of the effect of the initial induction condition.
- Statistical Analysis Reporting: ANOVA is mentioned, and significance letters are used in Table 2 and Table 3. However, the specific post-hoc test used for multiple comparisons is not stated. In Figures 1 and 2, significance between treatments is not directly indicated, only implied by trends described in the text. The statement "bud induction rate [...] of A. mangium [...] were not significantly different between different media" under -N seems contradicted by the visual differences in Figure 1 and the later selection of an "optimal" medium.
- Interpretation of "No Nitrogen Deficiency Symptoms": The authors state buds from -N media showed no obvious symptoms of N deficiency, only reduced stem length/leaf number and lighter green colour compared to F controls. Lighter green color is a classic symptom of N deficiency. The absence of severe symptoms might be due to sufficient endogenous N reserves in the large stem explants, carry-over N, or the relatively short duration (40 days for induction). Claiming N is "not essential" based on this observation might be an overstatement; it appears less critical for initial bud break and short-term growth under these specific conditions, but long-term effects are unknown.
- Rooting Rate Interpretation: The study concludes "no significant difference" in the highest rooting rates achieved between buds originating from -N vs. F media. However, comparing only the peak rates achieved under different hormone combinations (Tx 8 vs Tx 12 for A. mangium; Tx 4 vs Tx 3 for A. auriculiformis) is potentially misleading. A direct comparison of rooting success for buds from Tx 8-(-N) vs Tx 8-(F) under the same optimal rooting condition for each species would be more informative about the effect of the induction medium itself.
- Figure Quality: Figures 1 and 2 use 3D bar charts which can obscure data and make comparisons difficult; 2D bar charts or line graphs would be clearer. Error bars are not shown, limiting assessment of variability.
- Table Formatting: Table 2 confusingly pairs -N and F treatments within the same row number, making direct comparison difficult. Presenting -N and F results in separate columns for each base medium/hormone combination would be much clearer. The use of double letters (e.g., "72.6ab") for significance in Table 2 is unusual and unexplained. Table 3 uses single letters, adding inconsistency.
- Language: While generally understandable, there are some grammatical errors and awkward phrases (e.g., "vary due to different media", "free pests and diseases", "rinse them off", "conducted a rooting experiment ... which provided important conditions" ).
- Mechanism Discussion: The discussion acknowledges that N deficiency affects stem length/leaf number and speculates on physiological adaptation. However, it lacks depth in exploring how the plants cope. Potential mechanisms like utilization of stored N in the stem explant, potential N fixation by endophytic bacteria (common in legumes like Acacia, though unlikely significant in non-sterile culture without specific inoculation), or simply a very low N requirement for the initial bud break phase are not adequately discussed.
- Comparison with Literature: The discussion cites studies on browning and general responses to N deficiency in other species. However, it lacks comparison with other studies attempting low-N or N-free tissue culture, or studies investigating the specific N requirements of Acacia spp. in vitro. Is this finding truly novel across plant tissue culture, or specific to Acacia?
- Sustainability Argument: The study claims importance for the "sustainable development of tissue culture technology" by solving procurement issues for nitrates/ammonium nitrate. While addressing a practical constraint is valuable, the discussion doesn't consider potential trade-offs (e.g., slower growth, altered physiology of plantlets, need for N addition later) or the environmental/cost implications of using alternatives if growth is significantly hampered long-term. The non-sterile approach, if truly viable, would have major sustainability implications (reduced energy/autoclave use), but its validity is questionable.
The manuscript presents preliminary evidence that initial axillary bud induction in A. mangium and A. auriculiformis can occur on MS-based media lacking the primary N sources (NH₄NO₃, KNO₃), with A. auriculiformis even showing a 100% induction rate on an optimized -N medium. This finding addresses a practical challenge related to chemical procurement. However, the study suffers from significant limitations: the non-standard non-sterile culture method requires strong validation, the definition and interpretation of "nitrogen deficiency" need refinement, statistical reporting and data presentation are unclear, and the mechanistic discussion is superficial. Claiming nitrogen is "not essential" seems premature based on the presented data. Major revisions are needed to address the methodological concerns (especially sterility), improve data presentation and statistical reporting, and provide a more nuanced interpretation and discussion before the findings can be considered robust and widely applicable.
Reviewer 3 Report
Comments and Suggestions for Authors
Introduction
Include in the Background articles that report the in vitro propagation of plant species under nitrogen deficiency conditions to present the state of the art in this research topic.
Material and Methods
Leave a space between the number and the measurement parameter, for example: 0.8 g instead of 0.8g; 3 min instead of 3min, and so on.
Results and Discussion
Compare the results obtained with those reported by: Y. Lu, L. Huang & H. Wang (2024) Acacia mangium × A. auriculiformis micropropagation in a non-sterile environment, Australian Forestry, 87:1, 17-25, DOI: 10.1080/00049158.2023.2298553
References
Write the names of the authors of the articles in upper and lower case letters
Update citation: DOI: 10.1080/00049158.2023.2298553
Round 2
Reviewer 2 Report
Comments and Suggestions for Authors
The authors have implemented the necessary corrections and revisions, and the manuscript is suitable for acceptance in its current form.
Author Response
Thank you for taking the time to provide constructive comments and feedback on this article.